# The functional organization of excitation and inhibition in the dendrites of mouse direction-selective ganglion cells

Varsha Jain[†], Benjamin L Murphy-Baum[†]*, Geoff deRosenroll, Santhosh Sethuramanujam, Mike Delsey, Kerry R Delaney, Gautam Bhagwan Awatramani

Department of Biology, University of Victoria, Victoria, Canada

**Abstract** Recent studies indicate that the precise timing and location of excitation and inhibition (E/I) within active dendritic trees can significantly impact neuronal function. How synaptic inputs are functionally organized at the subcellular level in intact circuits remains unclear. To address this issue, we took advantage of the retinal direction-selective ganglion cell circuit, where directionally tuned inhibition is known to shape non-directional excitatory signals. We combined two-photon calcium imaging with genetic, pharmacological, and single-cell ablation methods to examine the extent to which inhibition 'vetoes' excitation at the level of individual dendrites of direction-selective ganglion cells. We demonstrate that inhibition shapes direction selectivity independently within small dendritic segments (<10μm) with remarkable accuracy. The data suggest that the parallel processing schemes proposed for direction encoding could be more fine-grained than previously envisioned.

*For correspondence:
bmbaum@uvic.ca

[†]These authors contributed equally to this work

Competing interests: The authors declare that no competing interests exist.

## Introduction

Neural computations often rely on interactions between excitatory and inhibitory synaptic inputs (*Cafaro and Rieke, 2010*; *D'amour and Froemke, 2015*; *Denève and Machens, 2016*; *Rupprecht and Friedrich, 2018*; *Wehr and Zador, 2003*; *Xue et al., 2014*). While excitation can drive action potential firing, inhibition serves as an opposing force to gate excitatory activity (*Grienberger et al., 2017*; *Isaacson and Scanziani, 2011*; *Koch et al., 1982*; *Lovett-Barron et al., 2012*; *Muñoz et al., 2017*; *Murayama et al., 2009*; *Poleg-Polsky et al., 2018*; *Ranganathan et al., 2018*). Such interactions between excitation and inhibition (E/I) form the basis for logical 'AND-NOT' computations, in which activity is only propagated when excitation is present but inhibition is not (*Barlow and Levick, 1965*; *Major et al., 2008*; *Schachter et al., 2010*; *Sivyer and Williams, 2013*; *Wilson et al., 2018*). The direction-selective ganglion cell (DSGC) circuit in the retina is a classic example of a circuit that performs AND-NOT computations: objects in the visual scene that move over a DSGC's receptive field in its 'preferred' direction drive excitation onto its dendrites that trigger action potential firing, but for motion in the opposite or 'null' direction, excitation is 'vetoed' by coincident GABAergic inhibition (*Figure 1A–B*). The spatial scale over which such computations take place could have important impacts on the computational capacity of neurons (*London and Häusser, 2005*), but has not been investigated in intact circuits.

It is well-established that direction-selective (DS) GABAergic signals in DSGCs arise from the distal varicosities located on the radiating dendrites of presynaptic starburst amacrine cells (SACs; *Taylor and Smith, 2012*; *Vaney et al., 2012*; *Wei, 2018*). SAC varicosities are more effectively depolarized during centrifugal motion (from the soma to the dendritic tip), and thus the orientation of a given SAC dendrite provides an approximation of its preferred direction (*Euler et al., 2002*). In a seminal study combining physiological recordings with serial block-face electron microscopy,

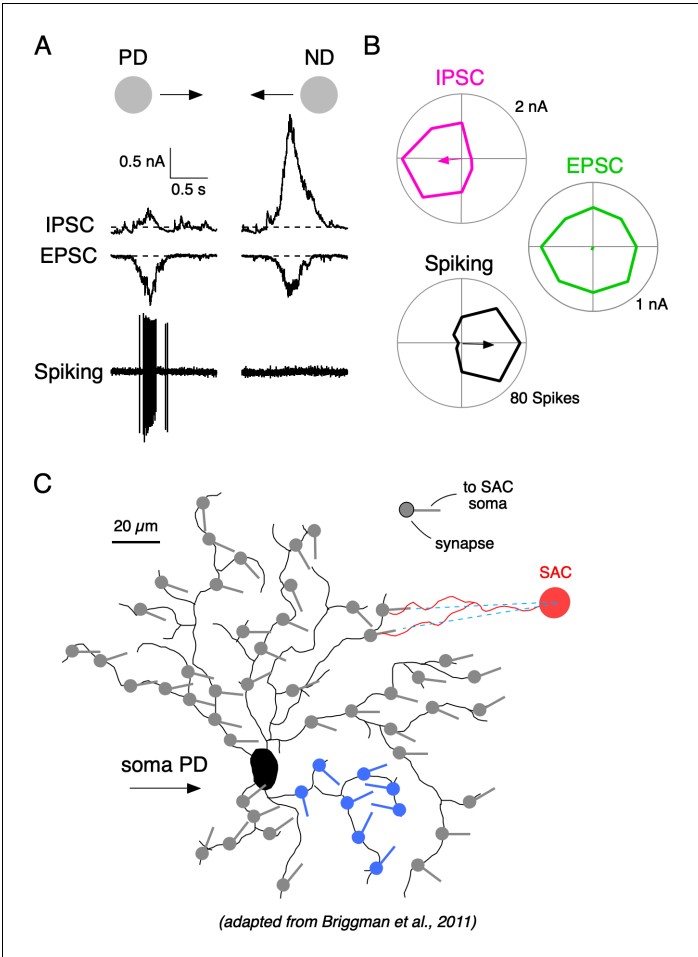

**Figure 1.** Synaptic mechanisms underlying direction selectivity in DSGCs. (**A**) Spiking, IPSCs, and EPSCs recorded from the soma of an ON-OFF DSGC in response to a positive contrast spot (200 µm diameter; 500 µm/s) moving in the DSGC's preferred (PD) or null direction (ND). (**B**) Polar plots of the peak amplitudes of the IPSC, EPSCs, and spiking responses evoked by spots moving in eight directions. Arrows indicate the vector sum angle, and its radius indicates the strength of the tuning. (**C**) Reconstructed DSGC overlaid with vectors representing the orientation of some of the presynaptic starburst amacrine cell (SAC) dendrites, which supply inhibition to it (adapted from SBEM reconstructions from *Briggman et al., 2011*; see their Figure 5). The orientations of SAC dendrites can predict the PD of DSGCs, because SACs release GABA maximally for stimuli moving from their soma towards their dendritic tips. The known anatomy predicts variable and sometimes disorganized local tuning in DSGCs (blue region). The online version of this article includes the following source data for figure 1:

**Source data 1.** Currents and spiking responses in DSGCs.

*Briggman et al. (2011)* demonstrated that SAC dendrites with similar orientations—and therefore similar directional preferences—tend to make synapses with the same type of DSGC (*Figure 1C*). This asymmetric wiring explains how individual DSGCs are biased to receive GABAergic inhibition in response to certain directions of motion but not others, which ultimately shapes their DS spiking responses (reviewed by *Vaney et al., 2012*; *Wei, 2018*). It should be noted that SACs also provide symmetrical excitatory cholinergic signals to DSGCs, which appear to be mediated by a diffuse synaptic mechanism that makes them insensitive to specific wiring patterns (*Briggman et al., 2011*; *Brombas et al., 2017*; *Lee et al., 2010*; *Sethuramanujam et al., 2017*). The resulting spatial offsets in the GABAergic and cholinergic connectivity produce E/I timing differences that contribute to the overall tuning properties of DSGCs (*Hanson et al., 2019*; *Koch et al., 1982*; *Taylor et al., 2000*; *Torre and Poggio, 1978*).

Although the overall tuning properties of DSGCs are evident from both their anatomical connectivity with SACs (*Briggman et al., 2011*) and somatic inhibitory currents (*Figure 1*), the nature of the DS information at the level of individual dendrites is not clear. A closer inspection of the orientation of presynaptic SACs shows that their dendritic orientation can change systematically throughout a DSGC's dendritic field and can be disorganized in particular dendritic subfields (*Figure 1C*; see dendrites highlighted in blue). Thus, the anatomy predicts that the inhibitory input to DSGCs will be heterogeneous, and result in frequent inaccuracies in local tuning in dendrites. However, interpreting the wiring at the scale of individual dendrites is difficult for several reasons.

First, in the original study, only a limited number of SACs were reconstructed (*Briggman et al., 2011*), so the differences in connectivity throughout the DSGC dendritic arbor may arise from a sampling bias. Second, $Ca^{2+}$ imaging studies suggest that there is significant variance in the directional tuning of individual SAC varicosities with similar dendritic orientations (*Morrie and Feller, 2018*), which is exacerbated by large trial-to-trial variability (*Ding et al., 2016*; *Poleg-Polsky et al., 2018*). Finally, it is also important to note that direction selectivity not only depends on the strength of SAC output, but also on its relative timing with excitation (*Cafaro and Rieke, 2010*; *Hanson et al., 2019*), so only examining the anatomy provides an incomplete picture. Thus, direct functional measurements from DSGC dendrites are required to ascertain the properties of subcellular direction selectivity.

To this end, the first set of dendritic patch-clamp recordings carried out in rabbit ON DSGCs provided invaluable insights into the local organization of inputs (*Sivyer and Williams, 2013*). Similar to reports in ON-OFF DSGCs (*Oesch et al., 2005*; *Trenholm et al., 2014*), this study directly demonstrated that DSGCs fire TTX-sensitive dendritic spikes that are directionally tuned. Critically, their data suggest that dendritic spike initiation is driven by localized excitatory activity, and that dendritic spike suppression requires inhibition to act in close proximity to the site of excitatory input (*Sivyer and Williams, 2013*). Thus, the glutamate, acetylcholine (ACh), and GABA that contribute to dendritic E/I in DSGCs must be coordinated both spatially and temporally in order to produce accurate DS information in dendrites.

Previous modeling estimates based on the natural branching patterns and theoretical voltage length constants of ganglion cell dendrites support the idea that DSGCs integrate information over 'subunits' (up to 50–100 µm diameter) that are electrically isolated from each other but relatively isopotential within (*Koch et al., 1982*; *Schachter et al., 2010*). If these dendritic subunits integrate synaptic inputs independently, the appropriate levels of E/I must be delivered to each subunit in a temporally precise manner in order to generate accurate DS responses locally (*Sivyer and Williams, 2013*). Here, we use 2-photon $Ca^{2+}$ imaging to measure how direction selectivity is expressed locally in ON-OFF DSGC dendrites, the spatial extent of independence of those signals, and how that relates to our expectations from previous anatomical and functional studies. The data suggest that independent and accurate DS information is present within small dendrite sections, indicating that synaptic inputs may be processed over a finer spatiotemporal scale than previously thought.

## Results

### Evaluating local synaptic activity in DSGC dendrites

ON-OFF DSGCs were identified based on their extracellular spiking responses to positive contrast spots moving in eight different directions (200 µm diameter, moving at 500 µm/s). To monitor activity in individual dendrites, DSGCs were loaded with $Ca^{2+}$-sensitive and -insensitive dyes (Oregon Green BAPTA-1 and Sulforhodamine 101, respectively) through a patch electrode (*Figure 2A–B*). Activity was typically imaged in the ON arbors of these bistratified DSGCs, where signals tended to be stronger than those observed in the OFF arbors, likely because the stimulus was a bright spot on a dark background. To isolate local activity, a voltage-gated sodium channel ($Na_V$) blocker was included in the electrode solution (2 mM QX-314; note, low concentrations were used to avoid significant $Ca^{2+}$ channel blockade; *Talbot and Sayer, 1996*), or in some cases 1 µM TTX was included in the bath solution. If $Na_V$ activity was left intact, the dendritic $Ca^{2+}$ responses would reflect the neuron's global activity due to the propagation of dendritic spikes or back-propagation of somatic spikes (*Oesch et al., 2005*; *Yonehara et al., 2013*), whereas our objective was to assess local synaptic activity. With $Na_V$ blocked, the dendritic $Ca^{2+}$ signals arise from $Ca^{2+}$ influx through both

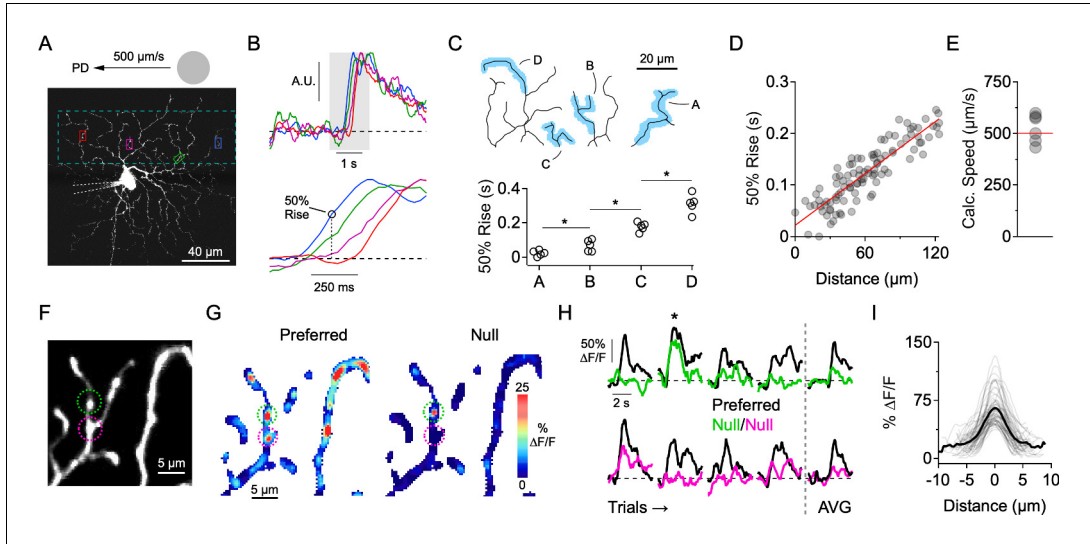

**Figure 2.** Independent activity in dendritic branches in the presence of Na$_V$ blockers. (**A**) 2-photon image of the ON-stratifying dendritic arbor of an ON-OFF DSGC, which was filled with a Ca$^{2+}$ indicator (Oregon Green Bapta-1, 200 µM), and a Na$_V$ channel blocker (QX-314, 2 mM) through a patch electrode. The dotted box indicates the imaging area. (**B**) Top, Normalized Ca$^{2+}$ signals for the four regions of interest (ROIs) demarcated in **A**. Bottom, expanded view of the shaded region, showing clear differences in onset time at the different ROIs. (**C**) 50% rise time from each of the four dendritic branches outlined in the above reconstruction. Asterisks indicate p<0.05, t test. (**D**) In an another cell, 50% rise time of the Ca$^{2+}$ signals versus lateral distance along the stimulus trajectory for many small ROIs that were placed throughout the dendritic tree. The red line is a linear fit to the data. (**E**) The slope of the linear fit predicts the speed of the stimulus (524 ± 28 µm/s), which was nominally set to be 500 µm/s. This data set included DSGCs that were recorded with (n = 3) and without (n = 3) the presence of the NMDA receptor antagonist D-AP5 (50 µM). No differences were found in terms of the onset timing of the Ca$^{2+}$ signals, so the data sets were combined. (**F**) 2-photon image of a section of dendrites from an ON-OFF DSGC, with two ROIs demarcated. (**G**) Spatial map of the peak ΔF/F for preferred (left) and null (right) motion. (**H**) Left, Time-varying Ca$^{2+}$ responses over multiple trials extracted from the 2 ROIs in **F** (green and magenta). Right, Average signals over four trials. The ΔF/F maps in **G** are from the second trial, in which a strong Ca$^{2+}$ signal in the green ROI (top) was observed during null motion. (**I**) Line profiles of the ΔF/F across active dendritic sites during null motion. The solid line is the average line profile, and the thin lines are from individual ROIs. Gaussian fits to the line profiles had an average FWHM of 3.0 ± 1.2 µm (µ ± s.d., n = 61 ROIs from 7 cells).

The online version of this article includes the following source data for figure 2:

**Source data 1.** Calcium responses and rise time measurements.

voltage-gated Ca$^{2+}$ channels (Ca$_V$) and NMDA receptors (*Poleg-Polsky and Diamond, 2016*; *Sethuramanujam et al., 2017*), which we provide evidence for in several subsequent experiments (Figures 2–3 and 6).

The first indication that individual dendrites integrate their synaptic inputs independently came from examining the relative timing onsets of Ca$^{2+}$ signals evoked by spots drifting across the DSGC's receptive field in its preferred direction (PD, computed from the vector sum of spiking responses evoked by spots moving in eight directions; see Materials and methods). Preferred motion, in which synaptic activity is mainly excitatory (*Figure 1A*), evoked a wave of activity as the stimulus traversed the dendritic arbor. Ca$^{2+}$ signals were first observed at dendritic regions of interest (ROIs) located closest to where the moving stimulus entered the DSGC's receptive field. Sites positioned later in the stimulus trajectory had progressively delayed response onsets (*Figure 2A–D*). Using the response latency and relative position of the dendritic sites, the stimulus speed was estimated to be 524 ± 28 µm/s (n = 6), close to the nominal stimulus speed (500 µm/s; *Figure 2E*). The sequential activation of dendrites was also observed when NMDA receptors were blocked (50 µM D-AP5). Since the relative timing of the Ca$^{2+}$ responses was similar to control conditions, we combined the data sets (*Figure 2E*). During NMDA receptor blockade, it is likely that the light-evoked

$Ca^{2+}$ signals are primarily mediated by $Ca_V$ channels, indicating that individual dendrites are depolarized in sequence as the stimulus crosses the DSGC's receptive field.

The data suggest that the dendritic arbor is not acting as a single unit, but rather that individual dendritic branches process their synaptic inputs independently. This idea supports previous theoretical and experimental work suggesting that motion information is processed in parallel by DSGC dendrites (*Brombas et al., 2017*; *Koch et al., 1983*; *Koch et al., 1982*; *Oesch et al., 2005*; *Schachter et al., 2010*; *Sivyer and Williams, 2013*; *Torre and Poggio, 1978*). To examine whether different regions of the dendritic tree receive adequate levels of inhibition to veto local excitatory responses, we next examined responses evoked during motion in the null direction, where inhibition is the strongest. We found that $Ca^{2+}$ signals evoked by null stimuli were significantly weaker or absent in the majority of sites throughout the dendritic tree (*Figure 2F–G*). Since the overall strength of excitation is similar to that evoked by motion in the preferred direction (*Park et al., 2014*; *Yonehara et al., 2013*; *Hanson et al., 2019*), the lack of $Ca^{2+}$ signals indicates the presence of strong and coincident inhibition at each dendrite.

However, we did occasionally observe activity in isolated dendritic 'hot spots' (FWHM = 3.0 ± 1.2 µm; n = 61 sites in 7 cells) during null motion (*Figure 2G–I*). These $Ca^{2+}$ signals were not consistently observed over multiple presentations of the null stimulus at any given site, suggesting that they arise from probabilistic transmission rather than from the absence of inhibitory synapses in that local dendritic region (*Figure 2H–I*). Such highly compartmentalized $Ca^{2+}$ signals, which persist for hundreds of milliseconds, may be indicative of strong buffering mechanisms in dendrites (*Awatramani et al., 2007*; *Biess et al., 2011*). While this reinforces the notion that the $Ca^{2+}$ signals reflect local synaptic activity, the highly localized signals evoked during null motion also raise the possibility that E/I interactions occur over a fine spatial scale.

Highly independent compartmentalized signaling was also apparent in response to orthogonal motion (i.e. motion 90° to the preferred-null axis; *Figure 3*). In this direction, where E/I levels are more comparable, the peak amplitudes of the $Ca^{2+}$ responses exhibited considerable trial-to-trial fluctuations (*Figure 3B*). To quantitatively assess the independence of dendritic sites, we assessed how $Ca^{2+}$ signals between pairs of ROIs co-varied as a function of the cable distance between them. In order to avoid measuring stimulus driven correlations (*Figure 3—figure supplement 1*), we computed the residual $Ca^{2+}$ fluctuations for each ROI by subtracting the mean response from the individual trials, over at least 30 consecutive trials in which the average response was stable. Using these 'noise residuals', cross-correlations were carried out between each ROI and every other ROI within the imaging window. Only nearby ROIs were significantly correlated with each other, and noise correlations measured in neighboring ROIs were absent when trials were shuffled (*Figure 3C*). The correlation strength rapidly decayed over distance with a space constant of λ = 5.3 µm (*Figure 3D–E*; n = 6 cells), suggesting that $Ca^{2+}$ signals beyond this distance arise independently.

Several pieces of evidence suggest that the trial-to-trial variability in the light-evoked $Ca^{2+}$ signals arises primarily from synaptic activity rather than from technical noise sources (e.g. PMT shot noise, movement artifacts). First, trial-to-trial variability was significantly lower when $Ca^{2+}$ signals were evoked by directly depolarizing the DSGC through the patch electrode in the presence of synaptic blockers (50 µM DL-AP4, 20 µM CNQX, 10 µM UBP-310; *Figure 3F–G*). Second, the response variability to motion in the PD—in which inhibition is quite low—was significantly less than that of orthogonal motion. Third, the response variability during orthogonal motion was significantly reduced in mice where GABA release from SACs was selectively disrupted using the conditional deletion of the vesicular GABA transporter ($Slc32a1^{fl/fl}::Chat^{Cre}$; vGAT KO; *Pei et al., 2015*; *Figure 3G*; *Figure 3—figure supplement 2*). However, since inhibition helps compartmentalize dendritic responses (*Lovett-Barron et al., 2012*; *Poleg-Polsky et al., 2018*), it is possible that part of the decrease in variability in the vGAT KO results from integrating excitatory inputs over a larger area. Nevertheless, these results confirm that the response variability is largely synaptic in origin.

## Directional tuning properties of small dendritic segments

So far, our results show that individual dendritic branches process their synaptic inputs independently and respond to moving stimuli in a DS manner (*Figures 2–3*). To better illustrate how the tuning properties of small dendritic segments vary across the dendritic tree, we measured $Ca^{2+}$ responses evoked by spots moving in eight directions over the DSGC's receptive field. $Ca^{2+}$

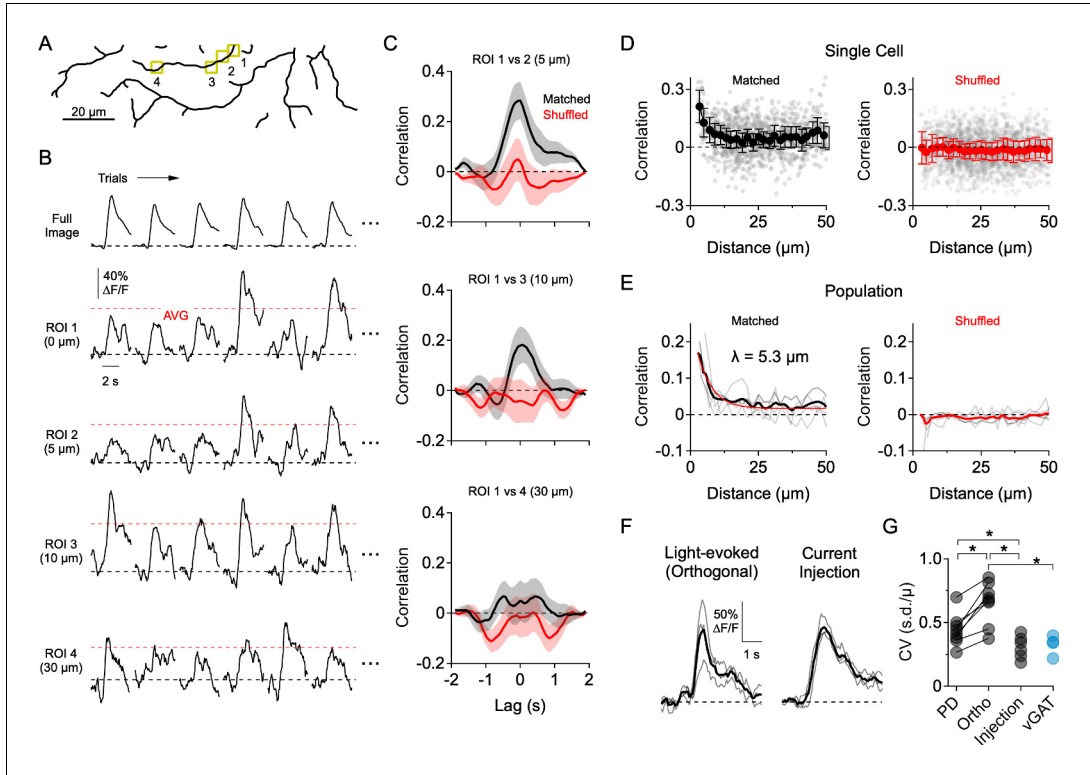

**Figure 3.** Dendritic Ca²⁺ signals form independently over small dendritic regions. (**A**) Reconstruction of the imaging area in the ON arbor of an ON-OFF DSGC. (**B**) Ca²⁺ signals over 30 consecutive trials in response to motion in a direction orthogonal to the preferred-null axis (only six trials are shown). Ca²⁺ signals were extracted from the ROIs highlighted in **A**, or from the entire arbor within the imaging window (Full Image). Red dotted lines indicate the average peak Ca²⁺ signal over all trials. (**C**) Cross-correlation of the mean-subtracted residual Ca²⁺ signals between ROI one and the other highlighted ROIs, each of which is at an increased distance away. Correlations were performed using matched trials (black) or shuffled trials (red). (**D**) Peak correlation versus distance between the ROIs for matched (black) and shuffled trials (red) in a single DSGC. Faded data points are individual correlation values, solid points are the average (± s.e.m.) over 2 μm bins. (**E**) Peak correlation versus distance for a population of 7 ON-OFF DSGCs for matched and shuffled trials. Faded lines are the average correlation versus distance data for single cells, bold lines are the population average. For matched trials, the thin red line is an exponential fit, with a space constant of λ = 5.3 μm. (**F**) Light-evoked Ca²⁺ signals from an example ROI in response to orthogonal motion compared to amplitude-matched Ca²⁺ signals evoked by current injection at the soma (measured in synaptic blockers: 50 μM DL-AP4, 20 μM CNQX, 10 μM UBP-310). Thin lines are individual trials; thick lines are the average. (**G**) Coefficient of variance (standard deviation divided by the mean) for the peak Ca²⁺ signals evoked by preferred motion, orthogonal motion, somatic current injection, and in response to motion in a mouse that has the vesicular GABA transporter knocked out of SACs (*Slc32a1*$^{fl/fl}$::*Chat*$^{Cre}$; vGAT KO). Asterisks indicate p<0.05, t-test.

The online version of this article includes the following source data and figure supplement(s) for figure 3:

**Source data 1.** Calcium responses, cross-correlation, and variability measurements.
**Figure supplement 1.** Stimulus-driven correlations between DSGC dendritic sites.
**Figure supplement 1—source data 1.** Stimulus correlation measurements.
**Figure supplement 2.** Dendritic Ca²⁺ signals are untuned when SAC inhibition is compromised.
**Figure supplement 2—source data 1.** Spiking responses, calcium responses, and tuning in DSGCs in the vGAT KO mouse.

responses extracted from small dendritic segments (3–4 μm in length) throughout the arbor were well-tuned for the direction of motion.

Dendritic nonlinearities mediated by Na$_V$ have been shown to play an important role in sharpening the overall tuning of DSGC output (*Oesch et al., 2005*; *Sivyer and Williams, 2013*). Indeed, the directional tuning of the somatic spiking was significantly narrower than that of the subthreshold

somatic voltage (measured with Na$_V$ activity blocked). Interestingly, the tuning width of the dendritic Ca$^{2+}$ signals was indistinguishable from that of the spiking response (*Figure 4A–C*), suggesting that the thresholding operations provided by dendritic Na$_V$ (*Oesch et al., 2005*; *Sivyer and Williams, 2013*; *Trenholm et al., 2014*) may be similar to those implemented by the nonlinearities associated with Ca$^{2+}$ influx.

Although the average PDs of nearby dendritic sites were similar, they did not necessarily covary over each stimulus set (eight directions per set), consistent with the idea that nearby dendritic sites are subject to different sources of synaptic noise. For example, when comparing the directions encoded by four neighboring regions on the same dendritic branch (*Figure 4D*), it can be seen that

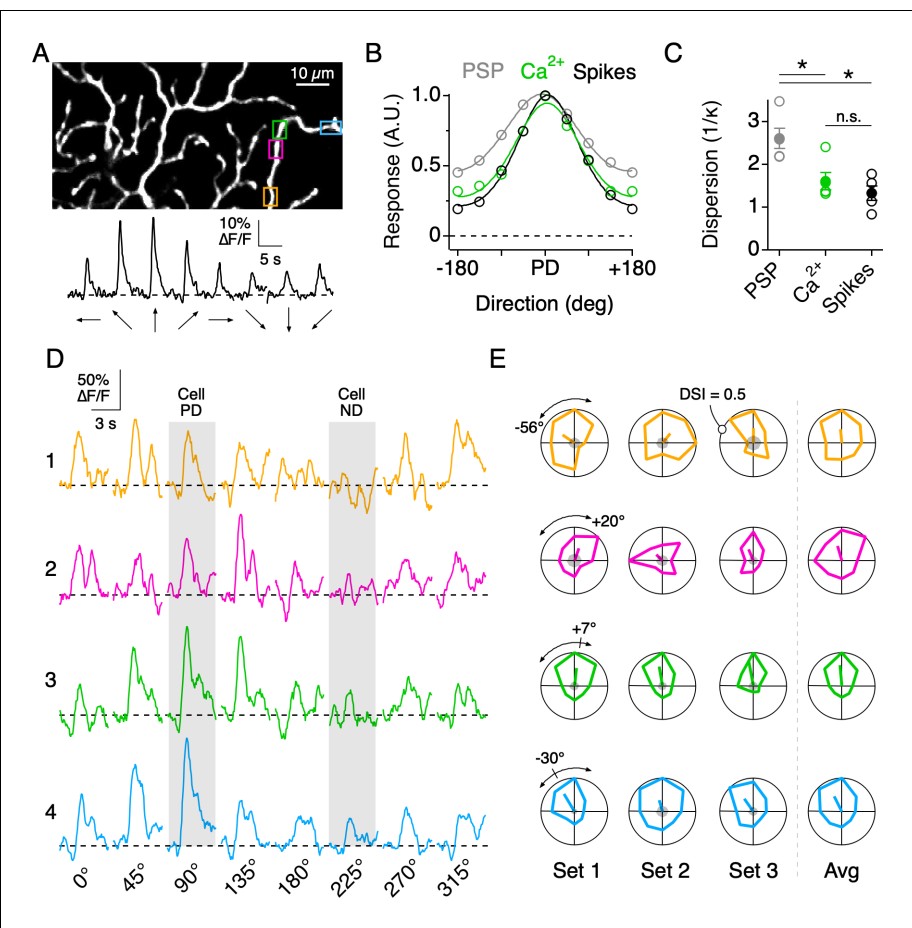

**Figure 4.** Directional tuning in DSGC dendrites in the absence of Na$_V$ activity. (**A**) Top, Imaging area of the ON arbor of an ON-OFF DSGC. Bottom, Ca$^{2+}$ signals over the entire imaging window in response to a positive contrast spot moving in eight different directions. (**B**) The average tuning curves of the spiking response, subthreshold somatic voltage (PSP), and dendritic Ca$^{2+}$ signals (n = 7 cells). Solid lines indicate Von Mises fits to the data. The PD is normalized to that of the spiking response. (**C**) The value 1/κ, extracted from the Von Mises fits, is a measure of tuning width, or angular dispersion, for circular data. The tuning curve from the somatic voltage was significantly broader than both the Ca$^{2+}$ signals and the spiking responses. Asterisks indicate p<0.05, t test. (**D**) Ca$^{2+}$ signals over a single stimulus set (one set = 8 directions) at four nearby ROIs on the same dendritic branch, as demarcated by the colored boxes in **A**. The PD and ND, determined using the extracellular spiking response, are indicated by the shading. (**E**) Polar plots of the peak Ca$^{2+}$ responses over consecutive stimulus sets, with the PD of each dendritic ROI indicated by the line extending from the origin, the length of which denoting the DSI, a measure of tuning strength (see Materials and methods). The arcs above the polar plot indicate the range in the preferred angle across these four sites on a single stimulus set. Right, the average polar plot for each ROI over the displayed stimulus sets.

The online version of this article includes the following source data for figure 4:

**Source data 1.** Calcium responses and tuning curves.

in the first stimulus set the PDs of ROIs 1 and 2 deviate by ~76°, while the PDs of ROIs 2 and 4 deviate by ~50°. In the next stimulus set, the deviations changed in sign and amplitude. Importantly, the baseline noise at each site before the stimulus onset was low compared to the peak amplitude of the $Ca^{2+}$ signals (*Figure 4E*, gray shading at the center of the polar plot). Note, for this and all subsequent analysis, only dendritic sites which responded to at least one direction of motion with a peak amplitude greater than three times the standard deviation of the baseline noise were considered.

Similar to the single branch described above, the directions encoded by dendritic sites distributed across larger fields of the arbor were tightly constrained around the DSGC's overall PD, as computed from its spiking response. The direction encoded by a given site rarely fell in the wrong directional hemisphere, even when computed over a single set of trials (9/353 sites in 7 cells, 2.5%). The overall variation in the PD from single stimulus sets (angular standard deviation, $\sigma_\theta$ = 52.8°; 353 ROIs from 7 cells) was reduced when three to five stimulus sets were averaged ($\sigma_\theta$ = 31.6°, *Figure 5A,C*). In addition, we found that the strength of the tuning, quantified using the direction selectivity index (DSI, ranging from 0 to 1; see Materials and methods), also varied among the population of dendritic sites (DSI = 0.19 ± 0.08; μ ± s.d.; 353 ROIs from 7 cells; *Figure 5B,D*). Interestingly, adjacent sites with similar PDs could have large differences in DSI, and vice versa, indicating that the directional tuning of each site arises independently (*Figure 5E*).

We also found that the peak magnitude of the $Ca^{2+}$ signals varied as a function of the DSI (*Figure 5F*; 353 sites from 7 DSGCs). In response to preferred motion, sites with a low DSI (<0.1) had significantly weaker $Ca^{2+}$ responses than those with a high DSI (ΔF/F = 0.60 ± 0.09 versus 1.02 ± 0.12, p<0.01, t-test, μ ± s.d.). However, the null responses at the low DSI sites were generally stronger than those observed at the high DSI sites (ΔF/F = 0.49 ± 0.08 versus 0.31 ± 0.05, p<0.01, t-test, μ ± s.d.). Despite their weak tuning, the population of sites with low DSI still computed the correct direction on average, deviating by 0.99 ± 52° from the PD computed from the spiking responses (n = 37 sites in 7 cells, μ ± s.d.). Together these data suggest that DS information is highly

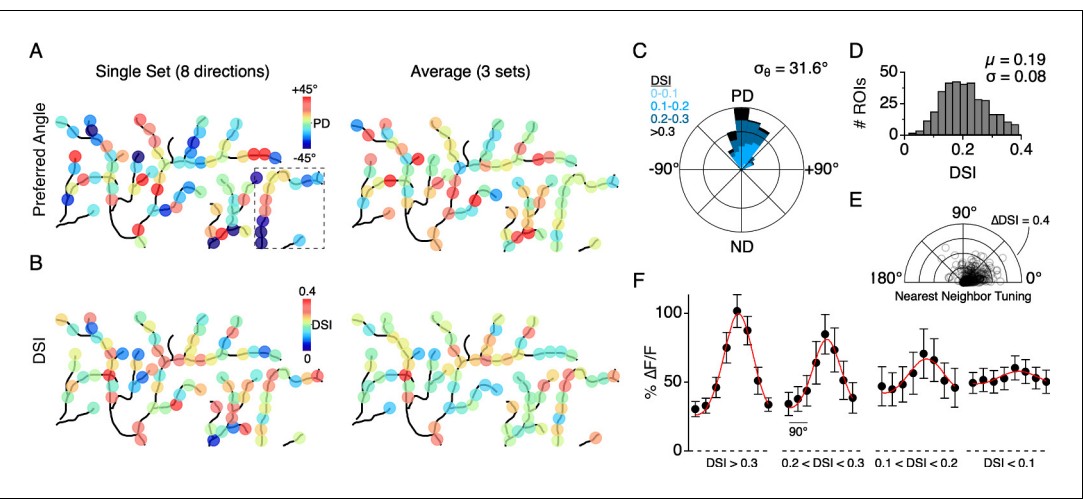

**Figure 5.** Distribution of directional tuning across the DSGC dendritic arbor. (**A**) Maps of the preferred angle of each dendritic ROI, as measured over a single stimulus set (left), or averaged over multiple sets (right). The boxed area denotes the region used for the analysis presented in *Figure 4*. (**B**) Same as in **A**, but for the DSI. (**C**) Rose plot showing the distribution of the average PD for ROIs across the dendritic arbors of 7 ON-OFF DSGCs, relative to the PD of each DSGC's spiking response (angular standard deviation, $\sigma_\theta$ = 31.6°). Colors indicate the proportion of ROIs within different ranges of DSI. (**D**) Distribution of DSIs for all dendritic ROIs (DSI = 0.19 ± 0.08, n = 353 ROIs from 7 cells). (**E**) Polar plot showing the difference in the DSI and PD between each ROI and its nearest neighboring ROI. (**F**) Average tuning curve for dendritic ROIs within different ranges of DSI. Note the amplitude of the null responses for weakly tuned sites (DSI <0.1) are on average larger than null responses measured for the strongly tuned sites (DSI >0.3).

The online version of this article includes the following source data for figure 5:

**Source data 1.** Vector angle and DSI measurements.

accurate throughout the dendritic arbor. Poorly tuned sites tended to have weaker responses, and therefore might not contribute to dendritic spiking as much as sites with stronger and more directionally accurate responses.

## The role of NMDA receptors in shaping the directional tuning of dendrites

Given that NMDA receptors mediate part of the light-evoked dendritic $Ca^{2+}$ signal, we wanted to test how they contribute to the overall dendritic tuning. Previous work suggests that the voltage-dependent properties of NMDA receptors allow them to act 'multiplicatively' on the DSGC membrane potential. That is to say, NMDA receptors amplify responses without changing the directional tuning properties (*Poleg-Polsky and Diamond, 2016*; *Sethuramanujam et al., 2017*). Consistent with this notion, we found that $Ca^{2+}$ responses measured in the presence of the NMDA receptor antagonist D-AP5 (50 µM) were weaker but still well-tuned for direction (*Figure 6A–B*). Under these conditions, the distribution of PDs (p=0.49, Angular Distance test; T [0.55]<$T_C$ [1.96], Watson-Williams test; *Figure 6C*) and DSIs (p=0.15, Kolmogorov-Smirnov test, *Figure 6D*) throughout the dendritic arbor were indistinguishable from that measured under control conditions. Thus, NMDA receptors appear to multiplicatively scale dendritic $Ca^{2+}$ signals, making them particularly useful for visualizing E/I interactions in the dendritic tree without disrupting them. These results are consistent with previous studies examining the role of NMDA receptors in shaping the somatic output from DSGCs, and directly confirm the prediction that multiplicative operations observed at the soma are also occurring in the dendrites (*Poleg-Polsky and Diamond, 2016*; *Sethuramanujam et al., 2017*).

In addition to NMDA receptors and $Ca_V$ channels, it is also possible that $Ca^{2+}$-induced $Ca^{2+}$ release (CICR) from internal stores amplified the measured $Ca^{2+}$ signals (*Grienberger and Konnerth, 2012*). In some trials, the $Ca^{2+}$ transients were significantly prolonged compared to the average, consistent with the idea that signals might be amplified by CICR (*Holbro et al., 2009*). However, these occurrences were rare, and since the peak $Ca^{2+}$ signals were measured within a brief window (200 ms), the delayed events would not have impacted our results. In addition, the fact that the directional tuning was not affected when $Ca^{2+}$ influx was reduced by blocking NMDA receptors also makes it unlikely that CICR plays a major role. Finally, other voltage-gated ion channels also

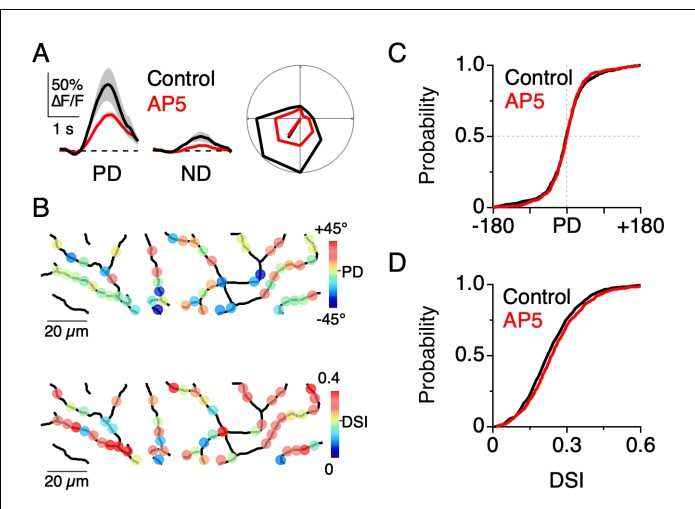

**Figure 6.** NMDA receptors amplify dendritic responses without altering direction selectivity. (**A**) Example $Ca^{2+}$ signals in response to preferred and null motion in control conditions (black) and in the presence of the NMDA receptor antagonist, D-AP5 (50 µM; red). Shading is ± s.e.m. Right, Example polar plot showing the directional tuning of dendritic $Ca^{2+}$ signals before and after NMDA receptor blockade. (**B**) Map of the PD (top) and DSI (bottom) for dendritic ROIs during NMDA receptor blockade. (**C**) Cumulative distributions of the PD for dendritic ROIs. (**D**) Cumulative distributions of the DSI for dendritic ROIs.

The online version of this article includes the following source data for figure 6:

**Source data 1.** Calcium responses, and dendritic tuning distributions.

likely play an important role in compartmentalizing signals (e.g. HCN, voltage-gated $K^+$ channels) but these were not further investigated.

## Dendritic nonlinearities promote dendritic independence

Synaptically-evoked $Ca^{2+}$ signals are expected to be more compartmentalized than dendritic voltage (*Wybo et al., 2019*; *Meier and Borst, 2019*; *Grimes et al., 2010*), since they arise from inherently nonlinear sources (voltage-gated $Ca^{2+}$ channels and NMDA receptors). Such nonlinearities are also expected to promote the independence of dendritic activity (*Grimes et al., 2010*), and could explain how nearby dendritic regions could have different directional tuning properties. To further test this idea, we constructed a multi-compartmental computational model of a reconstructed ON-OFF DSGC (*Figure 7*; *Hines and Carnevale, 1997*; *Poleg-Polsky and Diamond, 2016*; see Materials and methods). The model was driven by 177 inhibitory and excitatory inputs each, which integrated to produce somatic currents and spiking responses that were similar to those measured experimentally at the soma (*Figure 7B*). The inhibitory input at each site was directionally tuned in an identical manner, while the excitatory inputs were untuned. The changes in membrane voltage in response to a simulated edge moving across the dendritic arbor in different directions was recorded at each site. We tested how dendritic nonlinearities might affect directional tuning in the model by applying different thresholds to the dendritic voltage.

Since neurotransmitter release is stochastic, the relative strength of E/I for a given direction varied between sites and from trial-to-trial. Thus, the resulting directional tuning of local dendritic voltage signals (measured in the absence of $Na_V$ activity) differed between neighboring sites, even though all sites had similar average tuning properties. Similar to the $Ca^{2+}$ imaging data, the angle encoded at most sites varied about the DSGC's PD (*Figure 7B,C*). Since $Ca^{2+}$ signals rely on the activation of $Ca_V$ channels, which become steeply dependent on voltage above $-55$ mV (*Randall and Tsien, 1997*), we examined how dendritic tuning varies when responses are thresholded in this range. We found that systematically thresholding the voltage data from $-55$ mV to $-48$ mV led to an increase in both the angular variance ($\sigma_\theta = 25.4°$, $40.4°$, and $45.0°$ for $-55$ mV, $-50$ mV, and $-48$ mV thresholds, respectively) and the DSI (DSI $= 0.42$, $0.70$, and $0.80$ for $-55$ mV, $-50$ mV, and $-48$ mV thresholds, respectively; *Figure 7D*). When responses were strongly thresholded, even sites located on neighboring branches encoded different directions, indicating their functional independence (*Figure 7C*). The model data support the idea that dendritic nonlinearities could help promote compartmentalized directional tuning in dendrites.

## SAC input exerts local control of dendritic direction selectivity

So far, the major inference that we can draw from our results is that local E/I interactions play a dominant role in shaping DS signals in dendrites. As a more direct test of this hypothesis, we examined dendritic tuning after locally disrupting the inhibitory inputs arising from a limited set of SACs. To do so, we initially tried to apply local puffs of GABA receptor antagonists to block inhibition onto specific dendrites, but we found it difficult to keep the drug sufficiently localized. We also tried controlling GABA release from single SACs through a somatic patch electrode but found that the presence of light-evoked synaptic conductances made it extremely difficult to control SAC release. Finally, we resorted to single cell ablation methods using sharp electrodes to disrupt some of the inhibitory inputs mediated by SACs (*Jacoby et al., 2015*). In a mouse line in which SACs were genetically labeled (ChAT[Cre] x nGFP mouse line; *Vlasits et al., 2014*), 3–7 SACs with somas located on the null side of the DSGC's receptive field—which likely provide multiple strong GABAergic contacts to the DSGC—were targeted for ablation (*Figure 8A*). SAC ablation increased the spiking activity in the null direction and thus diminished direction selectivity, likely due to a decrease in inhibition from the ablated SACs (*Figure 8B*). Given that a typical DSGC receives synaptic contact from upwards of 20 SACs, our manipulation likely only disrupts a portion of the total number of GABAergic inputs throughout the arbor.

Notably, after SAC ablation we found that the directional tuning was highly heterogeneous within imaged regions of the DSGC dendritic arbor. Many dendritic sites were well-tuned for direction, similar to control conditions, while others were poorly tuned (*Figure 8C,E*). To abbreviate these otherwise lengthy experiments, stimuli were only moved along the preferred-null axis (note that this causes the reported DSIs to be higher than that measured using eight stimulus directions). The

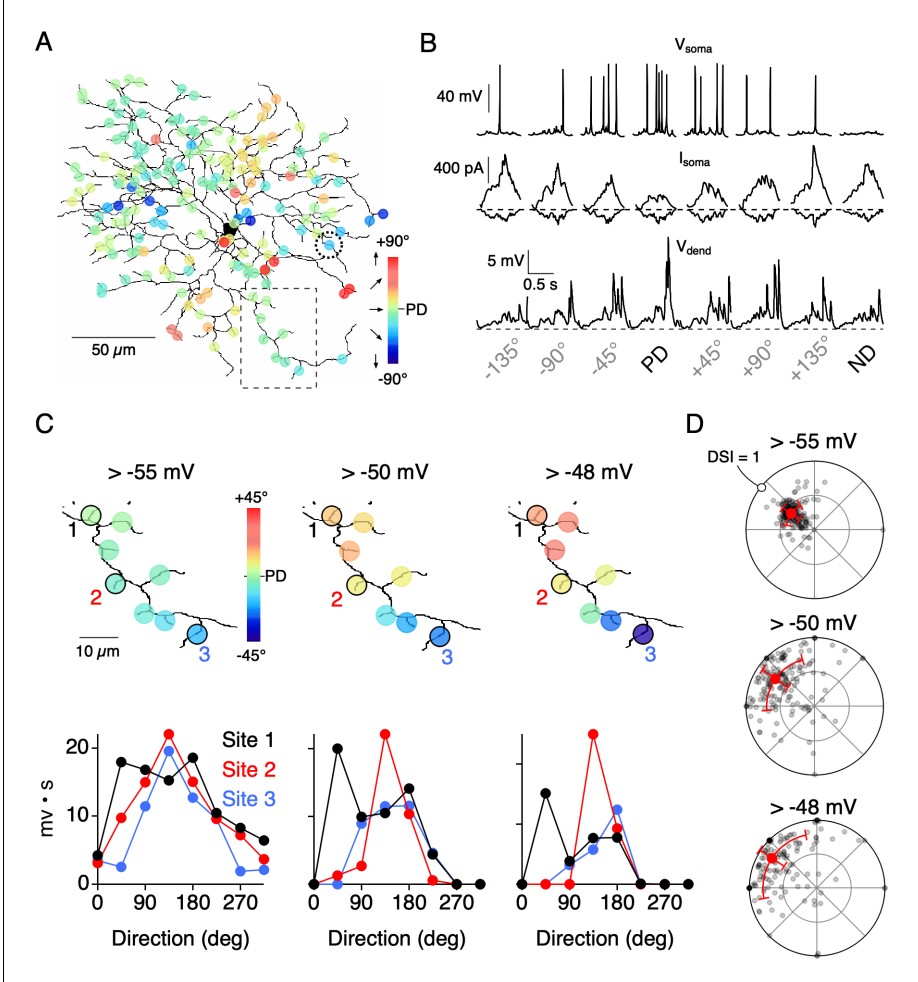

**Figure 7.** Threshold nonlinearities encourage dendritic compartmentalization. (**A**) Reconstruction of an ON-OFF DSGC used in a multicompartmental in NEURON (***Poleg-Polsky and Diamond, 2016***; see Materials and methods for details). Simulated responses to motion in eight directions gave rise to directional tuning at 177 dendritic sites. These responses were measured in the absence of Na$_V$ activity. (**B**) Top, Somatic voltage (V$_{soma}$) measured with Na$_V$ activity intact in response to 8 directions of motion. Middle and Bottom, somatic excitatory and inhibitory currents (I$_{soma}$), and voltage measured from a single dendritic site (V$_{dend}$; from site circled in **A**) in the absence of Na$_V$ activity. The PD of the integrated voltage responses (above a threshold of −55 mV) was used to compute the PDs shown in **A**. (**C**) Expanded view of the dendritic branch indicated by the boxed area in **A**, showing how the stringency of the voltage threshold (indicated above images) can affect the directional tuning in nearby dendritic sites. The integrated voltage at three sites is shown below. (**D**) Polar plots showing the distribution of PDs and DSIs measured over the dendritic tree, measured using different threshold values. Note, in the model all sites have the same average tuning, and the differences in tuning between sites only occur on single trial sets (eight directions) due to the stochastic nature of transmitter release.

The online version of this article includes the following source data for figure 7:

**Source data 1.** Dendritic voltage responses and directional tuning measurements.

overall DSI distribution was significantly different than that measured without SAC ablation (p=10$^{-16}$, Kolmogorov-Smirnov test; ***Figure 8E***). Consistent with the idea that SAC ablation results in a reduction in local inhibition, the null direction Ca$^{2+}$ signals at poorly tuned sites were large (***Figure 8D***). This contrasts with control conditions, in which poorly tuned sites had weak null and preferred responses (***Figure 5F***). Importantly, neighboring dendritic sites could have dramatically different DSIs after SAC ablation (***Figure 8D***). In some cases, we were able to measure the DSI before and after SAC ablation. Once again, clear changes in tuning strength occurred at many, but not all, sites throughout the arbor. Sites that were unaffected by SAC ablation were closely

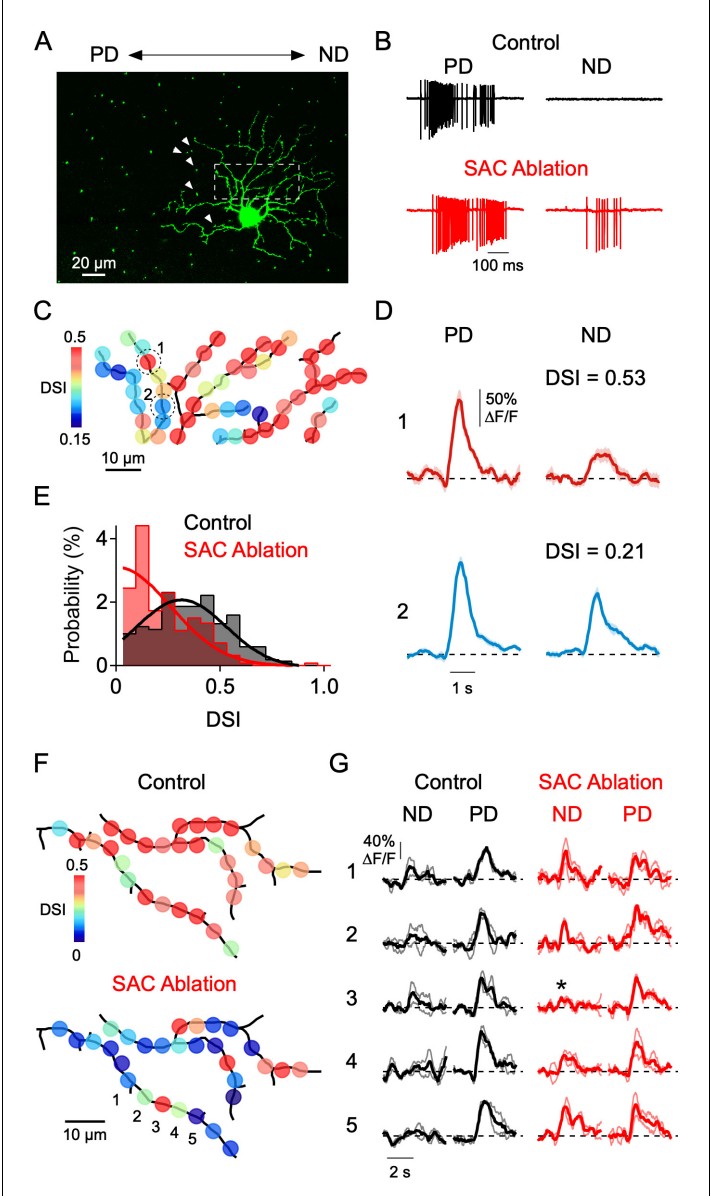

**Figure 8.** Local disruption of dendritic direction selectivity by SAC ablations. (A) Image showing the mosaic of SAC somas labeled with a nuclear GFP marker surrounding a DSGC filled with OGB-1. Arrowheads indicate null-side SACs that were individually targeted for electrical ablation with an electrode (see Materials and methods). (B) Spiking responses of a DSGC before (black) and after SAC ablation (red). (C) DSI map of the imaged area (dotted square in A), after 5 SACs were ablated from the null side of its dendritic arbor. (D) Average $Ca^{2+}$ signals (10 trials) in 2 ROIs (indicated in C) with strong and weak DSIs after SAC ablation. Shading is ± s.e.m. (E) Distribution of DSIs for a population of sites in 4 DSGCs after SAC ablation (red), compared with the distribution from DSGCs imaged under control conditions (black). Note, the DSI was calculated from only preferred and null stimulation, resulting in higher DSI values than those reported in *Figures 4–5*, where eight directions were used. Solid lines are Gaussian fits to the data. (F) Maps of the DSI before and after SAC ablation in the same DSGC. (G) Average $Ca^{2+}$ signals from five ROIs (1-5) in F, before (black) and after (red) SAC ablation. Thin lines are individual trials, and thick lines are the average response. Asterisk denotes a strongly tuned dendritic site flanked by poorly tuned regions. The online version of this article includes the following source data for figure 8:

**Source data 1.** Electrophysiology, calcium responses, and dendritic tuning measurements.

interspersed among strongly affected sites, suggesting that the inhibitory inputs to a given site have limited influence on the tuning of its neighbors (*Figure 8F–G*). These data demonstrate that local inhibitory inputs shape DS Ca$^{2+}$ responses within single dendritic branches.

## Discussion

### The functional organization of E/I in the dendritic arbors of DSGCs

Starburst dendritic varicosities are known to 'wrap-around' DSGC dendrites, and provide them with strong DS inhibition (*Briggman et al., 2011*). Functional imaging studies show that Ca$^{2+}$ signals in SAC varicosities are well-tuned for direction (*Ding et al., 2016*; *Euler et al., 2002*; *Koren et al., 2017*; *Poleg-Polsky et al., 2018*). However, the presynaptic Ca$^{2+}$ signals do not necessarily indicate the strength of the synapse or the extent to which GABA release shapes the postsynaptic response. To this end, our measurements of the tuning properties of postsynaptic DSGC dendrites provide several novel insights into the functional properties of inhibitory SAC synapses.

One of the central claims of this study is that SAC inhibition is highly directional and can act extremely locally (~5–10 μm scale). Inhibition vetoes local excitatory signals that appear non-directional, resulting in the production of small independent DS dendritic subunits throughout the DGSC's dendritic tree (*Figures 2–5*). Accurate DS information was present within each subunit even on a single set of trials. Given that there are at most 1–4 SAC synapses within each dendritic subunit (*Bleckert et al., 2013*; *Jeon et al., 2002*; *Sigal et al., 2015*), it is likely that the output of single SACs are graded according to direction, similar to their Ca$^{2+}$ responses (*Euler et al., 2002*; *Poleg-Polsky et al., 2018*). This was not entirely expected since transmitter release is nonlinearly related to Ca$^{2+}$ influx (*Dodge and Rahamimoff, 1967*). Support for these hypotheses came from the SAC ablation experiments, where we found that removing inhibition arising from a few SACs on the null side of the receptive field resulted in the loss of direction selectivity in small sections of the dendrite (~10 μm; *Figure 8*). After SAC ablation, robust DS and non-DS sites could be observed on the same dendritic branch, demonstrating the fine spatial scale over which inhibition can veto excitatory responses. It is also possible that some of the independence of DS subunits arises from variable levels of local excitation. However, since inhibition is thought to act as a shunt, its 'visibility' is expected to be more spatiotemporally constrained compared to excitation (*Koch et al., 1982*; *Rall, 1964*; *Torre and Poggio, 1978*; *Williams, 2004*), and may have a dominant effect on the local variance. Taken together, these data suggest that SAC varicosities provide reliable, localized, and systematic DS inhibition throughout the entire DSGC dendritic tree.

Although excitatory signals alone may drive depolarizations over larger dendritic areas, it is by no means uniform across the dendritic tree when a stimulus sweeps over the DSGC's receptive field—different dendrites are activated at different times (*Figure 2*). This makes the precise timing and placement of each inhibitory input from SACs more critical, since mistimed or misplaced E/I are not expected to strongly interact (*Hao et al., 2009*; *Koch et al., 1983*; *Liu, 2004*; *Lowe, 2002*; *Marlin and Carter, 2014*; *Müller et al., 2012*; *Müllner et al., 2015*; *Polsky et al., 2004*; *Takahashi et al., 2016*; *Williams, 2004*). Thus, the presynaptic networks must deliver the 'correct' amount of E/I to each dendrite for each given direction with temporal precision.

The tight spatiotemporal coupling of E/I is especially important for motion in the null direction, in which inhibition is tasked with fully preventing the generation of dendritic spikes. In response to null motion, we only occasionally observed Ca$^{2+}$ responses that were localized to small dendritic segments, suggesting that E/I mostly arrive at the same time and place at each point in the dendritic arbor (*Figure 2*). While the precise temporal characteristics of E/I cannot be determined by Ca$^{2+}$ imaging, in a computational model based on the morphology of DSGCs, we found that even E/I delays in the range of ~10 ms could significantly compromise the directional spiking output of the DSGCs (deRosenroll and Awatramani, unpublished data). It is possible that spatiotemporally coordinated E/I naturally arises from the co-transmission of GABA and ACh from SAC synapses. ACh is released from SACs of all dendritic orientations onto DSGC dendrites, and input from a single SAC is capable of eliciting dendritic spiking (*Brombas et al., 2017*). Thus, co-transmitting GABA from the same varicosity would ensure that during null motion, E/I is spatiotemporally synchronous. In other directions, coordinated E/I is less important, since full suppression of dendritic spiking is not required.

In addition to ACh, bipolar cells release glutamate onto DSGC dendrites as a second source of excitation. It is less clear how GABA effectively counters glutamatergic excitation though, since glutamate is expected to arrive slightly later than the GABAergic inputs during null motion due to the lateral offset of SAC receptive fields (*Hanson et al., 2019*; *Wei, 2018*). However, the glutamatergic inputs—which are largely mediated by NMDA receptors—may not need to be as finely coordinated with GABA since they rely on ACh-mediated depolarizations in order to open (*Brombas et al., 2017*; *Sethuramanujam et al., 2016*). Moreover, since inhibition is slow to decay it is expected to be significantly more tolerant of excitation that arrives late as opposed to that arriving early (*Schachter et al., 2010*). Since the apparent goal of null inhibition is to suppress spiking, why the retinal DS circuit drives null excitation in the first place is still not entirely clear. One advantage of non-directional excitation is that fewer excitatory neurons are required to drive the 4 types of ON-OFF DSGCs encoding different cardinal directions (*Vaney et al., 2012*).

Together, our data support a model of locally-generated direction selectivity, in which inhibition arrives spatiotemporally close to excitation as synaptic activity traverses the arbor in response to null motion. The requirements for tightly coordinated E/I may be less stringent for other directions of motion, in which full spike suppression does not occur. Although this hypothesis has existed for many decades (*Koch et al., 1983*; *Koch et al., 1982*; *Torre and Poggio, 1978*), our study is the first to provide detailed experimental data as to the spatial scale over which accurate directional information exists in DSGC dendrites.

## Functional and anatomical comparisons

Our $Ca^{2+}$ measurements are in general agreement with anatomical predictions that inhibition is applied throughout the DSGC dendritic arbor. That being said, the distribution of directional tuning that we observed was more homogeneous than expected given the anatomical and functional variance of presynaptic SACs (*Briggman et al., 2011*; *Poleg-Polsky et al., 2018*). We did not observe systematic shifts in the tuning across different dendritic sectors as suggested by the anatomy (*Figure 1*), and rarely observed sites that were tuned to the wrong directional hemisphere (*Figures 4–5*); 95% of the dendritic sites fell within ±31.6° of the DSGC's true PD (two standard deviations). This is a tighter distribution than can be estimated from the anatomical connectivity, in which roughly 10% of SAC inputs pointed in the wrong directional hemisphere (*Briggman et al., 2011*). However, predicting the precise directional preference of SAC dendrites from their anatomical orientation is a coarse and often inaccurate approximation, as demonstrated by recent studies (*Ding et al., 2016*; *Morrie and Feller, 2018*; *Poleg-Polsky et al., 2018*). It is also possible that in our study, sites with weaker $Ca^{2+}$ responses reflect some of the miswiring, since these sites tended to be poorly tuned. A further possibility is that the 5–10 μm integration area that our $Ca^{2+}$ measurements report from is large enough to contain several inhibitory synapses (*Bleckert et al., 2013*; *Briggman et al., 2011*; *Jeon et al., 2002*; *Sigal et al., 2015*). Thus, single inaccurate inhibitory inputs may be averaged with their immediate neighbors, resulting in more directionally accurate $Ca^{2+}$ signals.

## Implications for local dendritic integration and parallel processing

Dendritic subunits refer to regions of the dendritic tree that function semi-independently (*Koch et al., 1982*; *Schachter et al., 2010*). The spatial scale of DS subunits depends on several factors, including the branching patterns, the cable properties, and the expression of different voltage-gated channels (*Wybo et al., 2019*; *Meier and Borst, 2019*; *Grimes et al., 2010*). By contrast, *Barlow and Levick (1965)* defined a 'DS subunit' as the smallest subregion of a DSGC's receptive field over which motion could elicit a DS spiking response (*Barlow and Levick, 1965*; *Rivlin-Etzion et al., 2011*). However, motion over spatially restricted regions of a DSGC's receptive field may activate synaptic inputs over a much larger region of the dendritic arbor, owing to the spread of signals through laterally extending SAC dendrites. Thus, the Barlow-Levick DS subunits are only loosely related to the dendritic subunits described here, and thus both subunits define different aspects of the modular circuitry.

Although our data cannot speak to whether each dendritic site is capable of independently triggering $Na_V$-dependent spikes, several pieces of evidence suggest that only a few inputs are required. First, paired recording studies in rabbit retina show that dendritic spiking in DSGCs can be triggered by the cholinergic synapses arising from a single SAC, which may only contact a few DSGC

dendrites (*Brombas et al., 2017*). Second, similar to the sequential activation of Ca$^{2+}$ responses that we observed (*Figure 2*), the general location of dendritic spike initiation is coupled to the position of the visual stimulus over the dendritic arbor, suggesting that local synaptic activity triggers dendritic spikes as a visual stimulus traverses the receptive field (*Brombas et al., 2017*; *Sivyer and Williams, 2013*). Third, in our study we found that activity is highly localized to a few distributed hot spots during null motion, implying that the few spikes that are often observed during null motion are likely generated by only a few synapses (*Figure 2*). In other directions, dendritic spikes are likely formed in a hierarchical manner through cooperative depolarizations from several smaller dendritic segments (*Brombas et al., 2017*; *Sivyer and Williams, 2013*). Our work supports this hypothesis and extends it by showing that E/I within small dendritic segments hold highly accurate DS information. Locally accurate information may be a critical requirement for parallel processing of motion information, in which local synaptic activity can drive independent computations simultaneously at different locations in the dendritic arbor (*Schachter et al., 2010*). A compelling prediction is that small integrative subunits would allow DSGCs to constantly resample motion information over their receptive fields, permitting fast adjustments in their spiking response to new or changing directional information.

The local dendritic integration model that the present data support contrasts with global integration schemes that have also been implicated in direction encoding. For instance, a recent analysis of subcellular glutamate transients in ON DSGC dendrites suggests that the glutamatergic excitatory input comes from kinetically distinct bipolar cell types (*Matsumoto et al., 2019*). Here, optimal temporal summation during preferred motion occurs over the course of the DSGC's entire dendritic arbor. Further experimentation is required to determine whether global integrative mechanisms operate in different visual scenarios, or whether they work together with local mechanisms to reinforce decisions being made by individual dendrites.

## Conclusions

Since the theoretical studies of *Rall (1962)*, there has been broad interest in understanding the role of dendrites in neural processing (*Stuart and Spruston, 2015*). Being the receiving units of neural information, dendrites are well positioned to perform complex computations on their inputs that are critical for behavior (*London and Häusser, 2005*). Previous work has discussed single dendrites as being the basic computational units of the brain (*Branco and Häusser, 2011*; *Branco and Häusser, 2010*). Our data suggest that during natural patterns of activity, accurate DS information is present within small sections of the dendrites, supporting the possibility that single dendrites process complex visual information in parallel. Interestingly, recent evidence suggests that DS neurons in the visual cortex also rely on E/I interactions (*Wilson et al., 2018*), and their circuits may be organized with more local specificity than previously envisioned (*Scholl et al., 2017*). Whether single neurons in cortical circuits require high spatiotemporal E/I coordination in order to prevent errant dendritic spiking remains to be determined. The recent advent of tools to optically monitor dendritic excitation (iGluSnFR; *Marvin et al., 2013*) and inhibition (iGABASnFR; *Marvin et al., 2019*) paves the way for exciting future investigations of how inhibition and excitation shape direction selectivity and other neural computations carried out by diverse circuits in the brain.

# Materials and methods

**Key resources table**

| Reagent type (species) or resource | Designation | Source or reference | Identifiers | Additional information |
|---|---|---|---|---|
| Gene (*M. musculus*) | B6.129S6-Chat$^{tm1(cre)lowl}$/J | PMID: 21284986 | RRID: IMSR_JAX:006410 | |
| Gene (*M. musculus*) | B6.129-Gt (ROSA)26Sor$^{tm1Joe}$/ | PMID: 18395835 | RRID: IMSR_JAX:008516 | |
| Gene (*M. musculus*) | Slc32a1$^{tm1Lowl}$ | PMID: 19160495 | RRID: IMSR_JAX:012897 | |
| Gene (*M. musculus*) | Trhr-EGFP | PMID: 14586460 | RRID: MMRRC_030036-UCD | |
| Gene (*M. musculus*) | Hb9-EGFP | PMID: 12176325 | RRID: IMSR_JAX:005029 | |

## Animals

Experiments were performed using adult (P30 or older) mice of either sex: *C57BL/6J* (RRID: IMSR_JAX:000664). SACs were genetically accessed using the choline acetyltransferase (ChAT) Cre mouse line (*B6.129S6-Chat^tm1(cre)lowl/J*; RRID: IMSR_JAX:006410). Cre-dependent expression of nuclear-localized GFP in SACs was achieved by crossing the *Chat^Cre* with *B6.129-Gt (ROSA)26Sor^tm1Joe/J* (RRID:IMSR_JAX:008516). To reduce SAC GABA release the *Chat^Cre* line was crossed with floxed *Slc32a1^tm1Lowl* (also commonly referred to as *Vgat^fl/fl*; RRID: IMSR_JAX:012897). To target DSGCs the *Trhr-EGFP* (kindly provided by Dr. Marla Feller, UC Berkeley; RRID: MMRRC_030036-UCD) and *Hb9-EGFP* (RRID: IMSR_JAX:005029) lines were used in combination. Animals were housed in 12 hr light/dark cycles, in up to five animals per cage. All procedures were performed in accordance with the Canadian Council on Animal Care and approved by the University of Victoria's Animal Care Committee.

## Tissue preparation

Mice were dark-adapted for at least 45 min before being anesthetized with isoflurane and decapitated. Retinas of both eyes were extracted in standard Ringer's solution under a dissecting microscope equipped with infrared optics. Isolated retinas were laid flat, ganglion cell side up, over a pre-cut window in 0.22 mm membrane filter paper (Millipore). Mounted retinas were placed into a recording chamber and perfused with Ringer's solution (110 mM NaCl, 2.5 mM KCl, 1 mM $CaCl_2$, 1.6 mM $MgCl_2$, 10 mM glucose, 22 mM $NaHCO_3$) heated to 35°C and bubbled with 95% $CO_2$/5% $O_2$. Retinas were illuminated with infrared light and visualized with a Spot RT3 CCD camera (Diagnostic Instruments) through a 40x water-immersion objective on a BX-51 WI microscope (Olympus Canada).

## Visual stimulation

Visual stimuli were produced using a digital light projector and were focused onto the photoreceptor layer of the retina through the sub-stage condenser. The background luminance was measured to be 10 photoisomerisations/s (R*/s). Moving stimuli were designed and generated using a custom stimulus design and presentation GUI (StimGen) in either a MATLAB environment with Psychtoolbox or in a python environment with PsychoPy (https://github.com/benmurphybaum/StimGen; copy archived at https://github.com/elifesciences-publications/StimGen; *Murphy-Baum, 2020a*).

## Electrophysiology

DSGCs were identified using 2-photon imaging in mouse lines with fluorescent labeling or were identified by their extracellular spiking responses in wild type mice. Electrodes were pulled from borosilicate glass capillaries to a resistance 3–6 MΩ. For extracellular recordings, electrodes were filled with Ringer's solution. For voltage clamp recordings, electrodes contained the following in mM: 112.5 $CH_3CsO_3S$, 7.75 CsCl, 1 $MgSO_4$, 10 EGTA, 10 HEPES, 5 QX-314-bromide (Tocris). For $Ca^{2+}$ imaging current clamp recordings, electrodes were filled with the following in mM: 115 K-Gluconate, 7.7 KCl, 10 HEPES, 1 $MgCl_2$, 2 ATP-$Na_2$, 1 GTP-Na, five phosphocreatine, 2 QX-314, 0.2 Oregon Green Bapta-1, and 0.05 Sulphorhodamine 101. For some recordings, QX-314 was omitted and 1 μM TTX was included in the bath solution instead. Signals were sampled at 10 kHz and filtered at 2 kHz in a MultiClamp 700B amplifier (Molecular Devices). Analysis was performed using custom routines in MATLAB (Mathworks) and Igor Pro (Wavemetrics). The following pharmacological agents were added directly to the superfusion solution: DL-AP4 (50 μM; Tocris Bioscience), D-AP5 (50 μM; Abcam Biochemicals), UBP-310 (10 μM; Abcam Biochemicals), TTX (0.5–1 μM; Abcam Biochemicals), CNQX (20 μM; Tocris Bioscience).

## Calcium imaging

For $Ca^{2+}$ imaging experiments, DSGCs were patch clamped in current clamp mode and filled with the calcium indicator Oregon Green Bapta-1 (0.2 mM; ThermoFisher Scientific). For some experiments, a $Ca^{2+}$-insensitive red dye, Sulforhodamine 101 (50 μM; ThermoFisher Scientific), was also included to help visualize the dendritic arbor. After waiting 15–20 min for the indicator to fill the dendrites, the microscope was focused onto the ON-stratifying dendritic layer. The ON dendritic arbors were imaged because they typically evoked stronger $Ca^{2+}$ signals than the OFF arbor,

possibly because our visual stimulus was a positive contrast spot on a dark background. A dark background was preferable for our purposes because it limits exposure of the photomultiplier tubes to visual stimulus light, reducing stimulus artifacts and simplifying analysis of the $Ca^{2+}$ signals.

2-photon excitation was delivered using an Insight DeepSee$^+$ laser (Spectra Physics) tuned to 920 nm, guided by X/Y galvanometer mirrors (Cambridge Technology). Image scans were acquired using custom software developed by Dr. Jamie Boyd (University of British Columbia) in the Igor Pro environment (http://svn.igorexchange.com/viewvc/packages/twoPhoton/). The laser power was rapidly modulated using a Pockels cell (Conoptics), such that laser power was extinguished during mirror flyback phases but was turned on during data acquisition phases. A counterphase signal was used to turn on and off the projector LEDs during flyback and scan phases, respectively. This method prevents the bright projector light from contaminating the fluorescence signal from the $Ca^{2+}$ indicator, and also minimizes the time that the tissue is exposed to the laser.

Photomultiplier tube (PMT) single photon currents were converted to voltages with a decay time constant of approximately 250 µs. These voltages were digitally sampled at a rate of 40 MHz, integrated for one microsecond through a custom time-integrating circuit (designed by Mike Delsey and Kerry Delaney, University of Victoria). This circuit outputs an analog voltage which was then digitized at 1 MHz (PCI-6110, National Instruments) for image formation. This helped decrease fluctuations in signal amplitude that arise from the asynchronous arrival times of single photons during the one microsecond dwell time at each image pixel, thereby improving signal to noise at low light levels characteristic of sparse synaptic activity.

Dendritic $Ca^{2+}$ responses were extracted from small, 3–4 µm regions of interest (ROI). The ROI size was selected based on the typical spatial extent of isolated dendritic hot spots, and were made as small as possible while maintaining an acceptable SNR. Extracted signals were smoothed using a 2$^{nd}$ order Savitzky-Golay filter.

## SAC ablation experiments

ON SACs on the null-side of the DSGCs, within 50–150 µm of its soma, were identified and targeted for physical ablation in $Chat^{Cre}::nGFP$ mouse line. Approximately ~3–7 identified SACs were mechanically ablated by injecting 20 nA current for 10–15 s until the SAC cell membrane was ruptured (*Jacoby et al., 2015*). $Ca^{2+}$ signals in the ON arbor of the targeted ON-OFF DSGC were then measured in response to preferred and null motion. DSIs were calculated according to *Equation 6* (see *Quantification and Statistical Analysis*). To compare these values to previous control experiments, control DSIs were recalculated using only preferred and null responses, rather than using all eight recorded directions. In some experiments, $Ca^{2+}$ responses were measured in DSGC dendritic arbors before and after SAC ablation. In these experiments, DSGCs were loaded with OGB via single cell electroporation (*Ding et al., 2016*), and responses were measured in the presence of tetrodotoxin (0.5 µM TTX).

## Computational modeling

A multi-compartmental model was coded in the NEURON environment (*Hines and Carnevale, 1997*). 177 pairs of excitatory and inhibitory synaptic inputs were distributed across the dendritic arbor of a reconstructed ON-OFF DSGC (*Poleg-Polsky and Diamond, 2016*). Membrane capacitance and axial resistance were set to 1 µF/cm$^2$ and 100 Ω•cm respectively. Membrane channels and noise were modeled using a stochastic Hodgkin and Huxley distributed mechanism. Non-voltage-gated leak conductances were set to reverse at −60 mV. Active membrane conductances were placed at the soma, primary dendrites and terminal dendrites with the following densities in mS/cm$^2$ (soma/primary dendrites/terminal dendrites): sodium (150/200/30), potassium rectifier (35/35/25), and delayed rectifier (0.8/0.8/0.8). This enabled the model DSGC to process inputs actively via dendritic sodium spikes. Sodium and potassium conductances were blocked for voltage-clamp recordings.

The membrane potential of 700 compartments across the dendritic arbor were recorded in response to a simulated edge moving across the receptive field at 1 mm/s. Onset times of each synaptic site varied around a mean value determined by the time at which the simulated edge passed over it. Random selections from Gaussian distributions determined whether or not synaptic release of inhibition and/or excitation was successful. For inhibition, the limits defining successful or

unsuccessful release depend on the direction of motion, with more stringent limits during preferred motion when the probability of release is low, and broader limits during null motion when the probability of release is high. The release probability for excitatory synapses was held constant across all directions.

## Quantification and statistical analysis

All analysis and statistical comparisons were done using Igor Pro (WaveMetrics). All population data were expressed as mean ± standard deviation unless otherwise specified. Angular data were expressed as angular mean ±angular standard deviation ($\sigma_\theta$). The preferred direction and strength of the directional tuning was defined as the vector summation of the component responses for the eight stimulus directions, which was computed as:

$$R_x = \sum_{i=1}^{n} R_i \cdot \cos\theta_i \tag{1}$$

$$R_y = \sum_{i=1}^{n} R_i \cdot \sin\theta_i \tag{2}$$

$$R_t = \sum_{i=1}^{n} R_i \tag{3}$$

$$\vec{R} = \sqrt{R_x^2 + R_y^2} \tag{4}$$

$$\vec{\theta} = \tan^{-1}\frac{R_y}{R_x} \tag{5}$$

where $R_i$ is the response for the $i^{th}$ stimulus direction ($\theta$), $R_x$ and $R_y$ are the x and y components of the response, and $R_t$ is the summed total of all responses. $\vec{R}$ and $\vec{\theta}$ are the length of the vector sum and its angle, respectively. The direction selective index (DSI) is computed as the normalized length of the vector sum:

$$DSI = \frac{\vec{R}}{R_t} \tag{6}$$

DSI measurements range from 0 to 1, with one indicating a response to only a single direction, and 0 indicating an equal response to all directions. Typically, a DSI > 0.2 has been used as a threshold for strong direction selectivity, but this number will vary depending on how many angles are actually sampled. For instance, sampling only preferred and null motion results in artificially high DSI values. For 8-direction stimuli, a non-DS ganglion cell would be expected to have a DSI << 0.1, and is often very close to zero.

The angular standard deviation of a population of angles was calculated as:

$$\sigma_\theta = \sqrt{-2 \cdot \log(V)} \tag{7}$$

where

$$V = \sqrt{\left(\sum_{i=1}^{n}\cos\theta_i\right)^2 + \left(\sum_{i=1}^{n}\sin\theta_i\right)^2} \tag{8}$$

and $\theta$ is the $i^{th}$ angle in the population.

Statistical comparisons between mean angles were done using the Watson-Williams test, and comparisons between the angular variance of two populations were done using the Angular Distance test, which is a Mann-Whitney-Wilcoxon test on the angular distances of each sample from its mean angle. Comparisons between mean DSIs over trials were done using the Student's t test, while

comparisons of distributions of DSIs (e.g. among many dendritic sites) were done using the Kolmogorov-Smirnov test. All other paired comparisons were done using the Student's t test. All comparisons were done using two-tailed statistical tests. Differences were considered significant for $p < 0.05$.

## Data and software availability

The data analysis code is available at https://github.com/benmurphybaum/eLife_2020_Analysis (*Murphy-Baum, 2020b*; copy archived at https://github.com/elifesciences-publications/eLife_2020_Analysis). The visual stimulation software (StimGen) is freely available online (https://github.com/benmurphybaum/StimGen).

## Contact for reagent and resource sharing

Further information and requests for resources and reagents should be directed to, and will be fulfilled by, the Lead Contact, Gautam B Awatramani (gautam@uvic.ca).

## Additional information

### Funding

| Funder | Grant reference number | Author |
|---|---|---|
| Canadian Institutes of Health Research | 159444 | Gautam Bhagwan Awatramani |

The funders had no role in study design, data collection and interpretation, or the decision to submit the work for publication.

### Author contributions

Varsha Jain, Data curation, Formal analysis, Writing - review and editing; Benjamin L Murphy-Baum, Conceptualization, Data curation, Software, Formal analysis, Visualization, Writing - original draft, Writing - review and editing; Geoff deRosenroll, Data curation-Neuron Modeling; Santhosh Sethuramanujam, Data curation, Writing - review and editing; Mike Delsey, Resources, Electrical engineering support, and built custom hardware for the PMT amplifiers, and the time-integrated photon sampling circuits; Kerry R Delaney, Gautam Bhagwan Awatramani, Conceptualization, Writing - original draft, review and editing, Resources, Supervision, Funding acquisition, Methodology; Gautam Bhagwan Awatramani, Resources, Supervision, Funding acquisition, Methodology

### Author ORCIDs

Varsha Jain (iD) https://orcid.org/0000-0002-1620-4177
Benjamin L Murphy-Baum (iD) https://orcid.org/0000-0001-6746-3091
Geoff deRosenroll (iD) https://orcid.org/0000-0002-5431-2814
Gautam Bhagwan Awatramani (iD) https://orcid.org/0000-0002-0610-5271

### Ethics

Animal experimentation: All procedures were performed in accordance with the Canadian Council on Animal Care and approved by the University of Victoria's Animal Care Committee (Protocol 2016 (15).

### Decision letter and Author response

Decision letter https://doi.org/10.7554/eLife.52949.sa1
Author response https://doi.org/10.7554/eLife.52949.sa2

## Additional files

### Supplementary files

• Transparent reporting form

## Data availability
All data generated or analyzed during this study are included in the manuscript and supporting files. Source data files have been provided for all figures.

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
