## [Decision Letter]

**Acceptance summary:**

How the retina extracts the direction of motion of a visual object ranks as a cardinal problem of computation in neural circuits. Much has been revealed about this over the past half century, but the present paper offers an exquisitely detailed window on the underlying mechanisms in the fine neuronal dendrites of the retinal ganglion cell.

**Decision letter after peer review:**

Thank you for submitting your article "The functional organization of E/I in the dendritic arbors of retinal direction-selective ganglion cells" for consideration by *eLife*. Your article has been reviewed by three peer reviewers, including Markus Meister as the Reviewing Editor and Reviewer #1, and the evaluation has been overseen by Ronald Calabrese as the Senior Editor. The following individual involved in review of your submission has agreed to reveal their identity: William Grimes (Reviewer #3).

The reviewers have discussed the reviews with one another and the Reviewing Editor has drafted this decision to help you prepare a revised submission. Please aim to submit the revised version within two months.

Summary

Jain et al. explore the functional organization of excitation and inhibition (E/I) on dendrites of ON-OFF DS RGCs in mouse retina. This is an extremely well studied circuit, but several details of the computation remain to be understood. The key question in this study is whether imprecise directional tuning in the dendrites averages out at the soma or whether, instead, directional tuning is already precise in small dendritic segments. The authors argue for the latter, showing very precise tuning and suggesting independent synaptic input down to the scale of ~5-10 μm. The results are consistent with a biophysical picture in which (1) Ca transients in the DSGC are spatially confined on a scale <10 µm, and (2) synapses separated by ~10 μm receive input from different SAC dendrites, and (3) those SAC dendrites are tuned to within 30*°* (SD) of the same null direction.

Essential revisions

1) Framing of the study.

These observations about the spatial structure of SAC inputs to the DSGC are interesting and mostly consistent with expectation from the known anatomy. However, the authors frame the report in a very different light: as a study of spatially and temporally precise excitation-inhibition interactions within dendrites, for example: "which allowed us for the first time to directly infer the degree to which the presynaptic excitatory and inhibitory inputs to DSGCs are functionally organized at a subcellular level"; "suggest that presynaptic excitatory and inhibitory inputs are precisely organized"; "patterns of presynaptic activity are coordinated with enough precision to support computations based on fine scale E/I integration"; "suggests that the underlying excitatory and inhibitory inputs are highly coordinated throughout the DSGC's dendritic tree"; "underlie the high fidelity of excitation and inhibition"; "tight spatial and temporal E/I coordination".

These claims are only half-supported. In the conventional wisdom, the excitatory input is almost entirely untuned and direction selectivity results from the inhibitory inputs. All the interesting dynamics of integration that produce direction-selectivity have been accomplished by the SAC already. Consistent with this conventional picture, the present experiments primarily report on inhibitory inputs to the DSGC, rather than any precise interaction of excitation and inhibition. There are multiple indications in this work that the details of excitation matter little, other than offering a baseline signal that can be inhibited: First, the overall excitatory input to the DSGC is untuned (Figure 1A). Second, when the authors perturb the excitation by blocking NMDA receptors the dendritic tuning remains the same (Figure 4). Third, the experiments do not manipulate the relative timing of excitation. In fact, one suspects that the dendritic tuning would be identical if they supplied the excitatory input entirely through the recording electrode. By contrast, when the inhibition gets removed, by knockout of vGAT (Figures 5C,D, Figure 6G) or with SAC ablation (Figure 8), the fluctuations and the DS in the Ca response mostly disappear. So the report is primarily about inhibition, not any detailed functional organization between excitation and inhibition.

We recommend that the authors reframe the study to emphasize what has been learned about the spatial distribution of inhibitory inputs. In general the observations seem to accord remarkably well with prior expectations from the anatomy of SAC-DSGC connections and the literature on presynaptic direction tuning at that same synapse (e.g. Poleg-Polsky et al., 2018). This itself may be a satisfying conclusion, but any deviations from those expectations could be highlighted as well.

As an alternative the authors might actually test whether there is any high-fidelity coordination of excitation and inhibition on a small scale, but that will require different experiments: separate manipulation of excitation and inhibition, and releasing the block on the dendritic sodium channels to observe any threshold nonlinearities that process the summed signal. A conventional picture of dendritic integration would suggest that E and I signals get summed over distances much greater than 10 μm before a spike is initiated.

2) The study seems to focus entirely on directional encoding in the ON arbors of the ON-OFF RGC without a clear justiﬁcation or explanation as to why. The spike recordings show responses to the leading and trailing edges of to the moving spots, are the resultant calcium signals conﬁned to their respective arbors? Do OFF dendrites show motion signals on a similar spatial scale?

3) The manuscript often refers to "active conductances" being blocked when in fact only Na_V_ channels are blocked. Voltage-gated Ca^2+^ channels are essential for the measurement, and there is no discussion at all about possible roles of voltage-gated K^+^ channels, or other active channels in the dendrites, if they exist. Some explanation would be helpful.

4) Subsection “Independent synaptic processing within small dendritic segments”, Figure 6: This analysis is about noise correlations, namely the fluctuations trial-to-trial that occur under the identical visual stimulus. Those fluctuations may well have a different origin from the systematic signal that produces directional tuning. The difference between noise and signal correlations and their interpretations and possible origins should be brought out better.

5) Subsection “Dendritic nonlinearities promote dendritic independence”: The model in Figure 7 seems poorly constrained. It is not clear how many free parameters there are, how they were chosen, and how robust the main ﬁnding is over parameter space.

6) Discussion section: Expand the section that relates known circuit anatomy to function. For example from the SAC coverage factor one can estimate how many null-side SACs connect to each OODS, providing additional context for the ablation experiments. Dramatic change in DSI in some spots and not others suggests that each hotspot might correspond to output from a single SAC. Similarly, the length scale of noise correlations may be related to the axonal arbor of a single bipolar cell. Non-specialists might not fully appreciate the size difference between an OODS RGC and the bipolar cells that are providing its excitatory input.

7) Subsection “Functional versus anatomical characterization of the circuit”: "The directional tuning of dendritic sites was relatively homogeneous; 95% of the dendritic sites fell within 63° of the DSGC's true PD (two standard deviations). This is a much tighter distribution than can be estimated from the anatomical connectivity, which shows SAC dendrite orientations roughly equally represented across a 90° spread." Actually, the opposite seems to be the case and the anatomical connectivity sounds slightly tighter: a uniform distribution ranging over 90° has a standard deviation of 26°, less than the 31° reported here for dendritic tuning. Actually though, the close correspondence is another indication how well the observations in the present report accord with prior understanding from physiology and anatomy in this circuit.

---

## [Author Response]

Essential revisions1) Framing of the study.These observations about the spatial structure of SAC inputs to the DSGC are interesting and mostly consistent with expectation from the known anatomy. However, the authors frame the report in a very different light: as a study of spatially and temporally precise excitation-inhibition interactions within dendrites, for example: "which allowed us for the first time to directly infer the degree to which the presynaptic excitatory and inhibitory inputs to DSGCs are functionally organized at a subcellular level"; "suggest that presynaptic excitatory and inhibitory inputs are precisely organized"; "patterns of presynaptic activity are coordinated with enough precision to support computations based on fine scale E/I integration"; "suggests that the underlying excitatory and inhibitory inputs are highly coordinated throughout the DSGC's dendritic tree"; "underlie the high fidelity of excitation and inhibition"; "tight spatial and temporal E/I coordination".These claims are only half-supported. In the conventional wisdom, the excitatory input is almost entirely untuned and direction selectivity results from the inhibitory inputs. All the interesting dynamics of integration that produce direction-selectivity have been accomplished by the SAC already. Consistent with this conventional picture, the present experiments primarily report on inhibitory inputs to the DSGC, rather than any precise interaction of excitation and inhibition. There are multiple indications in this work that the details of excitation matter little, other than offering a baseline signal that can be inhibited: First, the overall excitatory input to the DSGC is untuned (Fig Figure 1A). Second, when the authors perturb the excitation by blocking NMDA receptors the dendritic tuning remains the same (Fig Figure 4). Third, the experiments do not manipulate the relative timing of excitation. In fact, one suspects that the dendritic tuning would be identical if they supplied the excitatory input entirely through the recording electrode. By contrast, when the inhibition gets removed, by knockout of vGAT (Figs.Figures 5C,D, Fig. Figure 6G) or with SAC ablation (Fig. Figure 8), the fluctuations and the DS in the Ca response mostly disappear. So the report is primarily about inhibition, not any detailed functional organization between excitation and inhibition.We recommend that the authors reframe the study to emphasize what has been learned about the spatial distribution of inhibitory inputs. In general the observations seem to accord remarkably well with prior expectations from the anatomy of SAC-DSGC connections and the literature on presynaptic direction tuning at that same synapse (e.g. Poleg-Polsky et al.et al., 2018). This itself may be a satisfying conclusion, but any deviations from those expectations could be highlighted as well.As an alternative the authors might actually test whether there is any high-fidelity coordination of excitation and inhibition on a small scale, but that will require different experiments: separate manipulation of excitation and inhibition, and releasing the block on the dendritic sodium channels to observe any threshold nonlinearities that process the summed signal.

A) We agree with the reviewers that the variations in Ca^2+^ signals that we observe on single branches likely arise from variations in SAC output. We now illustrate the response variations at an earlier stage (Figure 3) and state this explicitly in the Discussion section (“since inhibition is thought to act as a shunt, its “visibility” is expected to be more spatiotemporally constrained compared to excitation (Koch et al., 1982; Rall, 1964; Torre and Poggio, 1978; Williams, 2004), and may have a dominant effect on the local variance.”)

B) However, we respectfully disagree with reviewer’s opinion that excitation is global (i.e. the dendritic tuning would be identical if the excitatory input was supplied through the recording electrode).

First, previous work has shown that inhibition is unable to shape excitation when excitation is provided through a dendritic recording electrode. Sivyer and Williams, 2013, demonstrated that dendritic spikes driven by a dendritic current injection are not suppressed by null motion (their Figure 7B-C) because the timing/location of the current injection (excitation) and the inhibitory input are not well matched.Second, previous studies have shown that excitation does not act globally throughout the dendritic arbor of DSGCs. Using paired somato-dendritic recordings from ON DSGCs in rabbit, Sivyer and Williams, 2013, showed that the location of dendritic spike initiation is closely tied to the location of the visual stimulus (their Figure 1), indicating that dendritic spikes are triggered by excitatory activity local to the initiation site.Similarly, in our study, we found the Ca^2+^ signals arise in dendritic branches are initiated at different times according to the location of the visual stimulus, confirming that excitation is acting locally in different dendritic branches. This result was originally a supplemental figure, but based on the reviewer’s comments, we moved it to a main figure (Figure 2 in the revised manuscript). We also support this result with further statistical analysis and discussion in the main text.

C) If excitation is non global, then it follows that SAC inputs would need to provide inhibition that is closely tied to excitation in space and time. Support for this idea comes from several studies described below:

As mentioned about, dendritic spikes driven by a dendritic current injection are not suppressed by null motion because the timing/location of the current injection (excitation) and the inhibitory input are not well matched (Sivyer and Williams, 2013).Schachter et al., 2010 also demonstrated the requirement of well-timed E/I at a given location, using a computational model.Cafaro and Rieke, 2010, demonstrated that excitation and inhibition were most strongly coordinated on a fine temporal scale for motion in the null direction (i.e. the strength of the noise correlations was highest for null motion). Poor E/I *temporal* coordination caused spurious spiking in the null direction and decreased the fidelity of stimulus encoding at the DSGC soma.In our experiments, the tight *spatial* coordination of E/I is indicated by the sparsity of Ca^2+^ responses to null motion. If inhibition was placed at a distance, strong Ca^2+^ responses would be expected to be consistently triggered at the points of excitation, which we did not observe in control conditions. This suggests the E/I are spatiotemporally co-activated as the stimulus crosses the dendritic arbor (except for a few instances, see text for detail). This result is now detailed in Figure 2 of the revised manuscript.

D) Finally, the reviewers refer to conventional DS models in which “the excitatory input is almost entirely untuned and direction selectivity results from the inhibitory inputs”, and conclude that “all the interesting dynamics of integration that produce direction-selectivity have been accomplished by the SAC already”. However, given the requirement for spatiotemporal E/I coordination, we argue that the SAC DS output only represents one important aspect of the circuit.

In fact, in a recent study we found that abolishing DS in SAC dendrites leaves direction selectivity in the DSGC relatively intact. In the absence of DS SAC output the E/I timing differences alone can generate robust direction selectivity. Thus, the overall effectiveness of DS GABAergic inhibition in DSGCs depends *both* on the amplitude of SAC GABAergic output as well as on its relative timing with excitation (Hanson et al., 2019).The direction-dependent timing differences in E/I are a natural outcome of the differential connectivity of GABAergic/cholinergic SAC inputs to DSGCs (Brombas et al., 2017; Chen et al., 2016; Lee et al., 2010; Yonehara et al., 2011). Thus, multiple specializations of the SAC circuitry underlie DS tuning at the level of DSGC dendrites. In the revised manuscript, we describe how timing differences can shape direction selectivity in more in detail.

In summary, given the narrow spatiotemporal window over which E/I is expected to interact, we argue that the dendritic tuning is not likely to be identical throughout the tree if the excitatory input was supplied through the recording electrode, as suggested by the reviewers. Thus, the strength of inhibition for each direction needs to be adjusted for each dendrite according to the amount of excitation it receives (if certain dendrites receive excess excitation, their Ca^2+^ responses would be expected to be more poorly tuned).

Finally, the overall strength of DS inhibition that shapes dendritic Ca^2+^ signals in DSGC dendrites is determined by not only the strength of SAC output, but also how well coordinated it is with local excitatory signals at each dendrite, which is an important aspect of this study. While the precise temporal characteristics of E/I cannot be determined by Ca^2+^ imaging, in a computational model based on the morphology of DSGCs, we found that disrupting tight E/I coordination could significantly compromise the directional spiking output of the DSGCs (deRosenroll and Awatramani, unpublished data; see Author response image 1). Here, the data show that the somatic currents cannot necessarily predict the spiking response, since the underlying E/I interactions are happening in the dendrites. We would be happy to include these data in the main manuscript if the reviewers feel that it brings clarity to this issue.

**Author response image 1. sa2fig1:** Correlated excitation and inhibition is critical for accurate and reliable direction selectivity. These data are from a NEURON model (as described in the main manuscript) where the spatiotemporal correlation between the excitatory and inhibitory inputs was manipulated. Temporal correlation is the likelihood that, at a given dendritic site, that excitation and inhibition arrive at the same time. Spatial correlation is defined as the likelihood that, at a given site, if excitation arrives that inhibition will arrive at the same location. A, Top, EPSCs and IPSCs measured at the soma in correlated and uncorrelated states in response to 8 different simulated directions of motion. Na_V_ activity is blocked for these recordings. Middle, Somatic spiking in response to correlated E/I inputs. Bottom, Somatic spiking in response to uncorrelated E/I inputs. Somatic and dendritic Na_V_ were active for the spike recordings. Note, the somatic currents are roughly the same in correlated and uncorrelated conditions, but more null spikes arise in the uncorrelated case. B, Top, E/I ratio across direction (measured at the soma) is similar in the correlated and uncorrelated states. Bottom, Directional tuning of the spiking response in the correlated and uncorrelated states. Note the increase in null spikes during non-preferred motion for the uncorrelated state. C, Polar plot form and vector angles (lines) of the directional tuning in correlated and uncorrelated states. D, DSI as a function of spatial and temporal correlation strength. Here, the spatial and temporal correlation was modeled independently. E, Same as D, but for the angular standard deviation. * indicates statistical significance p < 0.05 (t-test)..

A conventional picture of dendritic integration would suggest that E and I signals get summed over distances much greater than 10 μm before a spike is initiated.

We agree with the reviewer. Precisely how inputs sum to generate dendritic spikes is an interesting question, but not the main issue that is being investigated in this study. Our data only emphasizes local postsynaptic E/I interactions as viewed through the Ca^2+^ signal nonlinearity, which gives us valuable insights into the presynaptic organization of the circuitry and the spatial dimensions over which direction selective information exists in DSGC dendrites.

We make this clear in the Discussion section (“Although our data cannot speak to whether each dendritic site is capable of independently triggering Na_V_-dependent spikes…”)

That being said, it is worth noting that the thresholding operation that triggers spiking may be quite similar to the thresholding operation associated with Ca^2+^ influx, since the directional tuning of the spiking response and the dendritic Ca^2+^ responses were significantly more similar to each other than they were to the subthreshold voltage (Figure 3C).

2) The study seems to focus entirely on directional encoding in the ON arbors of the ON-OFF RGC without a clear justiﬁcation or explanation as to why. The spike recordings show responses to the leading and trailing edges of to the moving spots, are the resultant calcium signals conﬁned to their respective arbors? Do OFF dendrites show motion signals on a similar spatial scale?

Thank you for pointing this out, we have provided more explanation in the main text as well as the Materials and methods section. In our experiments, we used bright spots moving over a relatively dark ambient background, which generally produces stronger ON than OFF responses. Therefore, we imaged the ON arbors in order to take advantage of the improved Ca^2+^ signals. However, there were a few instances where we did image the OFF arbors and the results were qualitatively similar.

3) The manuscript often refers to "active conductances" being blocked when in fact only NaV channels are blocked. Voltage-gated Ca2+ channels are essential for the measurement, and there is no discussion at all about possible roles of voltage-gated K+ channels, or other active channels in the dendrites, if they exist. Some explanation would be helpful.

We now specifically state Na_V_ activity is being blocked. We include a discussion on the possible role that other voltage-gated channels could play in the integration of synaptic signals.

4) Subsection “Independent synaptic processing within small dendritic segments”, Figure 6: This analysis is about noise correlations, namely the fluctuations trial-to-trial that occur under the identical visual stimulus. Those fluctuations may well have a different origin from the systematic signal that produces directional tuning. The difference between noise and signal correlations and their interpretations and possible origins should be brought out better.

We now present a more detailed section discussing the differences in noise and signal correlations and their possible origins. For further clarity, we now show the stimulus-induced correlations separately in a supplemental figure (Figure 3—figure supplement 1). We also point out the experiments demonstrating that the source of the noise is synaptic, and that it depends on inhibition being present (Figure 3G).

5) Subsection “Dendritic nonlinearities promote dendritic independence”: The model in Figure 7 seems poorly constrained. It is not clear how many free parameters there are, how they were chosen, and how robust the main ﬁnding is over parameter space.

The model used was adapted from Poleg-Polsky and Diamond, 2016, and we refer readers to this article for the details of the model parameters. The model itself isn’t being fit to data—it is only a simulation—so there aren’t free parameters in that sense. We do test different voltage thresholds during our analysis of the model data, which is the only parameter that is varied. This should be clear from the main text and Materials and methods section. In the context of the current work, the point we wanted to emphasize with the model is that Ca^2+^ signals are a nonlinear transformation of voltage, which will tend to encourage signal compartmentalization. While this has been shown to be the case in many systems (Grimes et al., 2010; Meier and Borst, 2019), we thought that showing this effect in the context of dendritic DS would help readers quickly grasp the concept.

6) Discussion section: Expand the section that relates known circuit anatomy to function. For example from the SAC coverage factor one can estimate how many null-side SACs connect to each OODS, providing additional context for the ablation experiments. Dramatic change in DSI in some spots and not others suggests that each hotspot might correspond to output from a single SAC. Similarly, the length scale of noise correlations may be related to the axonal arbor of a single bipolar cell. Non-specialists might not fully appreciate the size difference between an OODS RGC and the bipolar cells that are providing its excitatory input.

We thank the reviewers for bringing this up, and we have expanded our discussion of the anatomical details of the DS circuit in the Introduction section, as well as in the SAC ablation description. We agree that the dramatic changes in local DSI suggest that each hot spot may be from a single (or a few) SAC inputs. As for the length scale of noise correlations, this likely reflects a combination of factors, rather than simply the size of the axonal arbor of bipolar cells (20-50μm). For instance, a series of axon terminals from a single bipolar cell likely contact completely different dendritic branches on a DSGC, since they project over a 20-50 μm area. It is also important to note that excitation arises from both glutamate from bipolar cells and acetylcholine from SACs.

7) Subsection “Functional versus anatomical characterization of the circuit”: "The directional tuning of dendritic sites was relatively homogeneous; 95% of the dendritic sites fell within 63° of the DSGC's true PD (two standard deviations). This is a much tighter distribution than can be estimated from the anatomical connectivity, which shows SAC dendrite orientations roughly equally represented across a 90° spread." Actually, the opposite seems to be the case and the anatomical connectivity sounds slightly tighter: a uniform distribution ranging over 90° has a standard deviation of 26°, less than the 31° reported here for dendritic tuning. Actually though, the close correspondence is another indication how well the observations in the present report accord with prior understanding from physiology and anatomy in this circuit.

We thank the reviewers for pointing this out; the statement that the anatomical data shows a uniform distribution across 90° is unclear. While a uniform distribution over 90° does indeed result in a 26° angular standard deviation, this number doesn’t take into account the remainder of the distribution, which for the anatomy is much broader than what we’ve recorded in our calcium measurements. We made the original statement trying to emphasize the difference in distribution width, but now can see how it is misleading. To be clearer, we have altered this statement, and now also describe our counts of oppositely tuned sites to show that the Ca^2+^ data are more tightly bound to the preferred direction than the anatomy predicts.